# Toxicity Study on Crude Alkaloid Extracts of *Houttuyniae herba* Based on Zebrafish and Mice

**DOI:** 10.3390/molecules29051107

**Published:** 2024-03-01

**Authors:** Jing Liu, Yingxue Wu, Yanni Xu, Ying Han, Shuai Kang, Zhong Dai, Hongyu Jin, Feng Wei, Shuangcheng Ma

**Affiliations:** 1National Institutes for Food and Drug Control, Beijing 100050, China; liujing_zsm@126.com (J.L.);; 2Institute of Medical Biotechnology, Chinese Academy of Medical Sciences, Beijing 100050, China

**Keywords:** crude total alkaloids extract, *Houttuyniae herba*, mouse, zebrafish, acute toxicity

## Abstract

*Houttuyniae herba* has a long history of medicinal and edible homology in China. It has the functions of clearing heat and detoxifying, reducing swelling and purulent discharge, diuresis, and relieving gonorrhea. It is mainly distributed in the central, southeastern, and southwestern provinces of China. *Houttuyniae herba* has been designated by the National Ministry of Health of China as a dual-use plant for both food and medicine. Comprising volatile oils, flavonoids, and alkaloids as its primary constituents, *Houttuyniae herba* harbors aristolactams, a prominent subclass of alkaloids. Notably, the structural affinity of aristolactams to aristolochic acids is discernible, the latter known for its explicit toxicological effects. Additionally, the safety study on *Houttuyniae herba* mainly focused on the ethanol, methanol, or aqueous extract. In this study, both zebrafish and mice were used to evaluate the acute toxicity of the total alkaloids extracts from *Houttuyniae herba* (HHTAE). The zebrafish experiment showed that a high concentration (0.1 mg/mL) of HHTAE had a lethal effect on zebrafish embryos. Furthermore, the mice experiment results showed that, even at a higher dose of 2000 mg/kg, HHTAE was not toxic. In conclusion, HHTAE was of low safety risk.

## 1. Introduction

*Houttuyniae herba*, a traditional Chinese medicine with a long history of use in China, is derived from *Houttuynia cordata* Thunb. of the Saururaceae family. Its medicinal properties include heat-clearing, toxin-eliminating, swelling-reducing, pus-discharging, and stagnation-relieving effects [1,2,3]. Phytochemical analysis has revealed that its main constituents are volatile oils, flavonoids, and alkaloids, which exhibit various bioactivities, such as anti-inflammatory, antiviral, antitumor, immune modulation, antioxidation, and antibacterial effects [4,5,6,7,8]. The alkaloids isolated from *Houttuyniae herba* mainly include aporphine alkaloids and aristolactams. Aristolactams have been identified as a naturally occurring compound and are considered the main metabolites of aristolochic acids by nitro-reduction reaction in vivo. Due to the toxic effects of aristolochic acids, the safety of *Houttuyniae herba* has been a topic of concern [9,10,11]. In our previous study, 12 aristolochic acid analogues, including five aristolochic acids and seven aristolactams, were systematically analyzed. It was found that aristolochic acid I, aristolochic acid II, aristolochic acid IIIa, aristolochic acid IVa, 7-hydroxy aristolochic acid I, and aristolactam I were either not detected in *Houttuyniae herba* or were below the method detection limits [12]. Therefore, the focus should be on alkaloids in terms of the safety-related substances in *Houttuyniae herba*.

In recent years, zebrafish has received much attention due to its advantages of high efficiency, economy, and strong predictability. And it plays an increasingly important role in studies of drug safety and efficacy [13,14,15,16,17]. In this study, we firstly report the evaluation of the acute toxicity of crude total alkaloid extracts from *Houttuyniae herba* (HHTAE) on zebrafish. Additionally, a mice experiment was also conducted to further verify the toxicity.

## 2. Results

### 2.1. Zebrafish Experiment

#### 2.1.1. Embryo Toxicity

The death of wild zebrafish embryos was observed under a microscope when exposed to different concentrations of HHTAE. The mortality rate was lower than 20% when the administration concentration was not greater than 50 μg/mL, and the LC_50_ value was 66.55 μg/mL (Figure 1).

#### 2.1.2. Phenotypic Effects of Zebrafish Embryo Development

The acute toxicity of HHTAE to zebrafish embryos had a dose–effect relationship. With the increase in the concentration of the drug administered, malformations (including death) also increased. At the concentration of 25 μg/mL, there were obvious abnormal phenotypes including delayed development and undissociated chorion. The embryo had a shorter body length, and the head and eyes became smaller. Additionally, there were cardiac developmental defects, large pericardial cysts, and poor blood flow in the ileum. There was enlargement of the yolk sac and abnormal phenotypes such as trunk and tail curvature (Figure 2A). Statistical analysis showed a dose-dependent increase in the fetal malformation rate and mortality. When the concentration reached 100 μg/mL, the toxicity was mainly lethal, with an embryonic mortality rate of over 90% (Figure 2B). The young fish of 5 dpf showed dose-dependent abnormalities including enlargement of the pericardium and abdomen and short body length.

#### 2.1.3. Effects of HHTAE on the Body Length of Zebrafish

HHTAE induced a concentration-dependent shortening of the body length of zebrafish larvae when compared with the normal control group (Figure 3).

### 2.2. Mouse Experiment

#### 2.2.1. Single Dose of 500 mg/kg and 2000 mg/kg

Both male and female mice survived without any abnormalities, and there was no mortality in either group. The diet and activity of the mice were normal, and there was no change in hair color. The liver and kidneys were normal under anatomical observation (see Appendix A). Compared with the control mice, there was no significant change in the body weight of both male and female administration mice. There was no significant difference in plasma ALT, AST, Urea, and Cre between the administration group and the control group using Student’s *t*-test (Figure 4A and Figure 5A). After administration, there were no visible changes in the organs of the male and female mice compared to the control. The H&E staining results showed that the liver and kidneys had no significant pathological changes after administration of 50 mg/kg (Figure 4B). Also, the liver showed no significant pathological changes after the administration of 2000 mg/kg (Figure 5B). However, the kidneys showed that the vacuolized epithelial cells in the renal tubules of both male and female mice were slightly more severe compared to the control group, and there were slightly more detached epithelial cells in the lumen than in the control group (Figure 5B).

#### 2.2.2. Continuous Administration of 2000 mg/kg, Twice a Day for 7 Days

Both male and female mice survived without any abnormalities, and there was no mortality in either group. The diet and activity of the mice were normal, and there was no change in hair color. The liver and kidneys were normal under anatomical observation (see Appendix A). Compared with the control mice, there was no significant change in the body weight of both male and female administration mice. There was no significant difference in plasma ALT, AST, Urea, and Cre between the administration group and the control group using Student’s *t*-test (Figure 6A). Compared with the control mice, the spleen of the male and female administration group mice became larger, and there were no visible changes in other organs (Figure 6B). The H&E staining results of the liver showed the infiltration of inflammatory cells and slightly increased vacuolar-like changes in the liver of both male and female mice after administration (Figure 6B). The H&E staining results of the kidneys showed no significant pathological changes (Figure 6B). The results of the spleen H&E staining showed a slight increase in the proportion of white pulp in the spleen of male and female mice after administration (Figure 6B).

## 3. Discussion

### 3.1. Necessity of Zebrafish Toxicity Experiment

Before conducting the animal experiments, the hepatorenal cytotoxicity in normal murine hepatocytes AML-12, rat hepatocytes BRL-3A, normal human hepatocytes L-02, rat renal tubular epithelial cells NRK-52E, African green monkey kidney cells Vero, and human renal cortex proximal tubule epithelial cells HK-2 cell lines by cell counting kit-8 (CCK-8) assay on different extracts of *Houttuyniae herba* was determined. The results showed that HHTAE exhibited cytotoxicity to AML-12, BRL-3A, L-02, and NRK-52E. HHTAE displayed stronger cytotoxicity when compared with other extracts, and it displayed a level of cytotoxicity comparable to that of the aristolactam components [11].

The result indicated the need for further toxicity studies.

As is well known, a toxicity experiment in vitro is a preliminary screening method. Due to their characteristics of efficiency, economy, and good predictability, zebrafish models have been increasingly applied in drug toxicity and safety evaluation. In our study, zebrafish was preferred as the model animal for further toxicity evaluation.

### 3.2. Solubility of Aristolactam Components

For the zebrafish experiment, the sample needed to have good water solubility since the zebrafish was cultured in artificial seawater. However, aristolactams showed poor water solubility (less than 1 μg/mL). This limitation could impact the ability to conduct experiments over a wide range of concentrations, affecting the precision and reliability of the results. Therefore, the experiment was not further conducted on the aristolactam components.

### 3.3. Use of a Single Extract

The in vivo experiments were conducted only on the HHTAE extract due to the insufficient quantities of aristolactam compounds, and poor water solubility. The results reflected the property of the test single extract. Considering its similar hepatorenal cytotoxicity to aristolactam components [11], the result could reflect a certain trend of the aristolactams to some extent.

### 3.4. Differences between Animal Models

The zebrafish experiment exhibited that HHTAE was mainly teratogenic within the tested concentration range. Additionally, there was no accurate interpretation of toxic effects based on the acute toxicity results of zebrafish model animal. To further confirm the toxic effects, a classic rodent mouse model experiment was conducted. The zebrafish and mouse animal experiment result showed that there was a certain difference between both models. This could have been caused by the different animals themselves or the different administration methods.

## 4. Materials and Methods

### 4.1. Plant Materials and Preparation

The dried *Houttuyniae herba* (30 kg) was purchased from Chengdu City, Sichuan Province. It was identified as the dry aerial part of *Houttuynia cordata* Thunb., a plant from the Saururaceae family, by associate professor Shuai Kang (Institute for Control of Chinese Traditional Medicine and Ethnic Medicine, National Institutes for Food and Drug Control). First, 30 kg of *Houttuyniae herba* was pulverized and extracted by heating and refluxing for 2 h with 12 times the amount of ethanol, 3 times. The solvent was then recovered, and the extract was combined. After suspending the extract with water, the pH value was adjusted to 8–9 with ammonia water. Then, it was extracted with an equal volume of dichloromethane, evaporated to dryness, and the HHTAE (440 g) was obtained.

### 4.2. Instruments and Reagents

A METTLER XS105 electronic analytical balance (Mettler-Toledo, Zurich, Switzerland), Milli-Q water purification system (Millipore, Burlington, VT, USA), KQ-500DE numerical control ultrasound cleaning instrument (Kun Shan Ultrasonic Instruments Co., Ltd., Kunshan, China), and an Olympus SZ61 body microscope (Olympus, Tokyo, Japan) were used. The basic animal feed was provided by SiPeiFu (Beijing) Biotechnology Co., Ltd. (Beijing, China). The artificial seawater was prepared with reverse osmosis water and purchased from Tianjin Zhongyan Marine Bioscience Co., Ltd. (Tianjin, China).

### 4.3. Animals

The wild-type zebrafish strain AB (Zebrafish, Danio rerio, AB strain) used in the experiment was cultured in artificial seawater (28.5 ± 1 °C). Sixty-eight specific pathogen-free (SPF) Kunming (KM) mice, 2 weeks old, weighing approximately 22 g, and of both sexes (34 females and 34 males), were used in the present study. The mice were obtained from Beijing Weitong Lihua Experimental Animal Technology Limited Company (Beijing, China) (certificate number: SCXK (Jing) 2021-0011). The experiments were approved by the Animal Care Ethics Committee of the Institute of Medical Biotechnology, Chinese Academy of Medical Sciences, and performed in the standardized pharmacology laboratory. All mice were acclimatized for 7 days before the experiments. All animals were housed at 22 ± 2 °C and constant humidity (40–70%) under a 12 h light–dark cycle. Male and female mice were housed separately and individually in sterile polypropylene cages and fed with basic feed and cold boiled water. After administration, the feed and water were freely given.

### 4.4. Zebrafish Experiment

#### 4.4.1. Administration Dose

An appropriate amount of HHTAE was weighed and subjected to ultrasonic extraction for 30 min (power: 500 W, frequency: 40 kHz) after adding a certain volume of water. After filtering, the filtrate was concentrated, and some water was added to prepare a stock solution of about 20 mg/mL. It was stored at −20 °C and diluted in different proportions during the test. According to the pre-experimental results, the sample at the concentration of 0.5 mg/mL was used as the original solution. Before use, 10 mL was taken and centrifuged at 5000 rpm for 5 min, and the supernatant was taken as the mother liquor.

#### 4.4.2. Acute Toxicity Test

(1)Grouping and administration: Half-encapsulated embryos (6 hpf under standard conditions) were taken, and the administration group was treated with a series of concentrated HHTAE solutions of 2 mL on zebrafish embryos. Embryos of the same batch of zebrafish developed in normal feeding liquid were the normal control group. Each group of 30 embryos was developed under standard conditions.(2)Observation: Embryos of both groups were observed and recorded daily under a microscope, including the brain, eyes, heart, blood flow, trunk (notochord/neural tube, somite), embryo growth rate, and so on (photos were taken and recorded at any time when necessary). At the third day of development (3 dpf), the administration group embryos were washed with normal culture medium 3 times, and the normal culture medium was replaced. The phenotypes of zebrafish (after membrane removal) were photographed, and the malformations (including body length) and the number of dead/surviving embryos of the embryos were counted. Then, they continued to develop until the embryo reached the fifth day, and they were recorded every day until the end of the experiment. The experiment was repeated at least three times.(3)Data processing: Mortality and body length were expressed as mean ± standard deviation (Mean ± SD). The data of each administration group were compared with the normal control group, the statistical data were plotted, and the median lethal concentration (LC50) was calculated.

### 4.5. Mouse Experiment

#### 4.5.1. Administration Dose

A certain amount of HHTAE was weighed and dissolved in an appropriate volume of 0.5% carboxymethyl cellulose sodium (CMC-Na). The administration doses of 500 mg/kg and 2000 mg/kg were administered separately.

#### 4.5.2. Acute Toxicity Test

(1)Grouping and administration: A certain number of KM mice weighing approximately 22 g, half females and half males, were selected and divided into four groups: male administration group, male control group, female administration group, and female control group. The administration group and the control group were kept in separate cages. Both groups were administered HHTAE at 500 mg/kg and 2000 mg/kg, with 0.5% CMC-Na by gavage, separately.(2)Toxic reaction observation: Within 14 days of administration, the mice were observed for any toxic reactions. This included general indicators such as animal appearance, behavior, secretions, excreta, etc., animal mortality (time of death, pre-death reactions, etc.), and changes in animal weight (weighed once before administration and before being killed at the end of the experiment). All deaths and symptoms, and the starting time, severity, and duration time were recorded. The end point time was 14 days later, and the mice were euthanized.(3)Pathological examination: All animals underwent gross dissection at the end of the experiment, including those euthanized during the experiment, those who died, and those still alive at the end of the experiment. The samples of plasma, heart, liver, spleen, and kidney were taken for biochemical and histopathology examination.

## 5. Conclusions

In order to evaluate the safety of *Houttuyniae herba*, a study on the determination of aristolochic acid analogues and the toxicity of HHTAE based on mouse and zebrafish animal models was conducted. The component analysis study showed that aristolochic acid I, aristolochic acid II, aristolochic acid IIIa, aristolochic acid IVa, 7-hydroxy aristolochic acid I, and aristolactam I were not detected in the *Houttuyniae herba* samples [12]. The zebrafish experiment showed that the acute toxicity of HHTAE to wild zebrafish embryos was mainly teratogenic within the tested concentration range of 5~50 μg/mL, with a dose–effect relationship. The teratogenic effects primarily involved the development of the embryonic trunk, nervous system, cardiovascular, and visceral organs of zebrafish. A high concentration (0.1 mg/mL) of HHTAE had a lethal effect on zebrafish embryos. Additionally, the toxicity experiment conducted on mice demonstrated that even at a high dose of 2000 mg/kg, HHTAE was not toxic. Since the yield of HHTAE was about 1.5%, it could be concluded that *Houttuyniae herba* would be not toxic at a much higher dose. In conclusion, both the determination and toxicity research showed that *Houttuyniae herba* is of low safety risk.

## Figures and Tables

**Figure 1 molecules-29-01107-f001:**
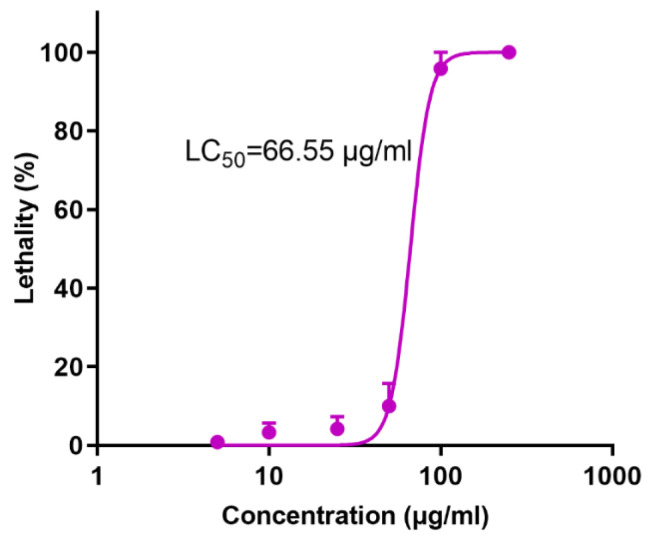
Dose–response curve of HHTAE causing death toxicity in zebrafish (3 dpf).

**Figure 2 molecules-29-01107-f002:**
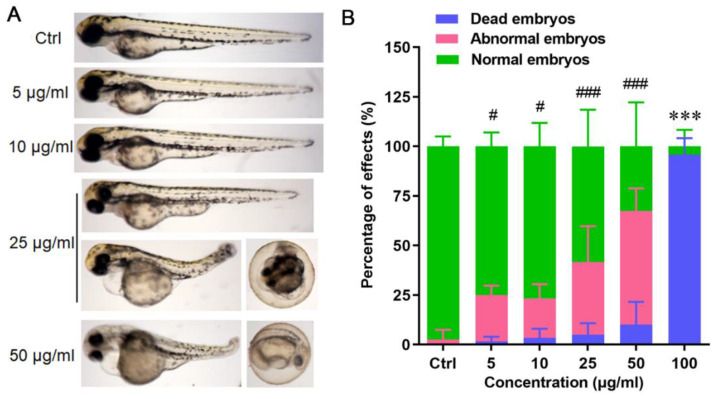
Effects of HHTAE on phenotypes of zebrafish. (**A**) HHTAE can cause malformation of zebrafish embryos (72 hpf). (**B**) Effects of HHTAE on the malformation rate and mortality of zebrafish embryos. #, ###: there was a significant difference in the abnormality rate between the treatment group and the normal group (#: *p* < 0.05; ###: *p* < 0.001); ***: there was a significant difference in the mortality rate between the administration group and the normal group (***: *p* < 0.001).

**Figure 3 molecules-29-01107-f003:**
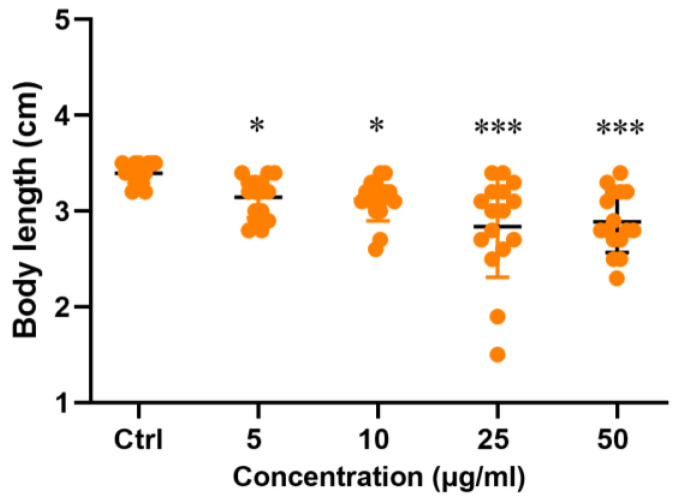
Effects of HHTAE on the body length of zebrafish (3dpf) (comparison of the body length of zebrafish in the HHTAE administration group with that of the normal group (*: *p* < 0.05; ***: *p* < 0.001)).

**Figure 4 molecules-29-01107-f004:**
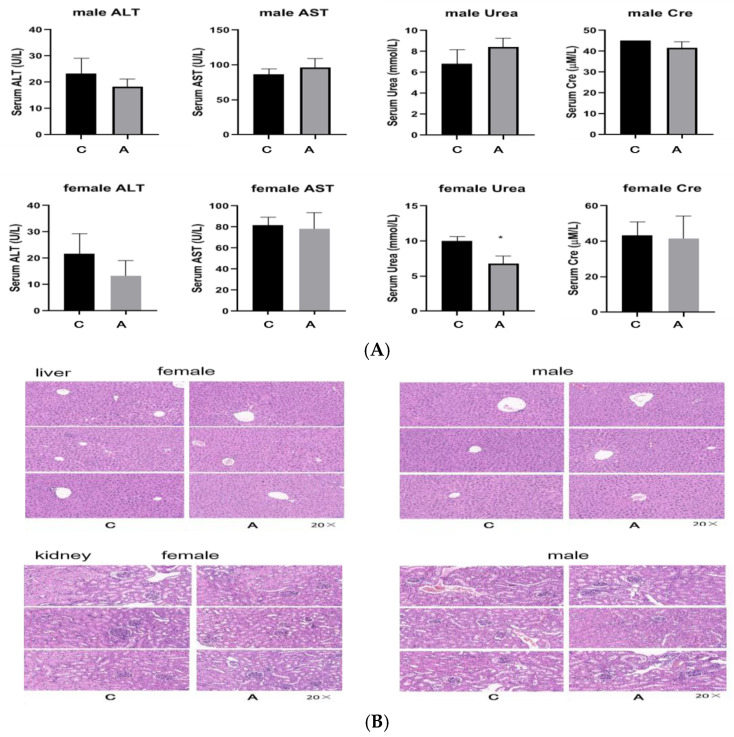
Results after the single dose of 500 mg/kg (C: control group, A: administration group, *: significance with *p* < 0.05). ((**A**) The numerical value of plasma liver and kidney function; (**B**) The pathological section of liver and kidney tissues).

**Figure 5 molecules-29-01107-f005:**
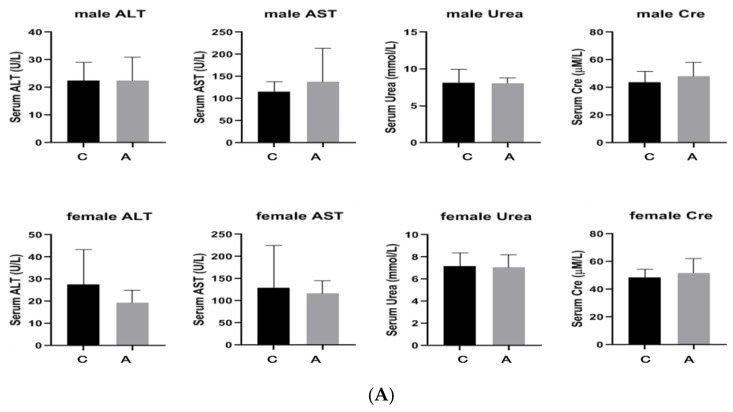
Results after the single dose of 2000 mg/kg (C: control group, A: administration group) ((**A**) The numerical value of plasma liver and kidney function; (**B**) The pathological section of liver and kidney tissues).

**Figure 6 molecules-29-01107-f006:**
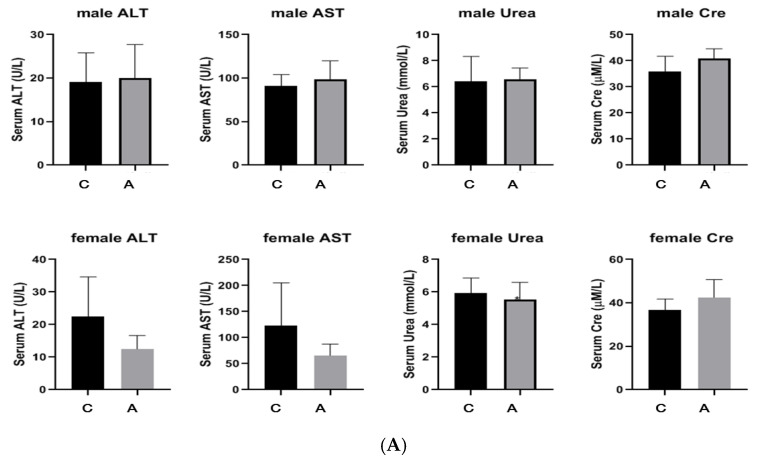
Results after continuous administration of 2000 mg/kg for 7 days (C: control group, A: administration group). ((**A**) The numerical value of plasma liver and kidney function; (**B**) The pathological section of liver, kidney, and spleen tissues).

## Data Availability

All data included in this study are available upon request by contacting the corresponding author.

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
