# Peer review of "Toxicity Study on Crude Alkaloid Extracts of Houttuyniae herba Based on Zebrafish and Mice"

_molecules, 2024, doi:10.3390/molecules29051107_

Round 1
Reviewer 1 Report
Comments and Suggestions for Authors
The manuscript by Liu et al. focuses on investigating the toxicity of alkaloid extracts from a traditional Chinese medicinal herb, Houttuynia Herba (HHTAE), on zebrafish and mice.
In the introduction, the authors reference their prior studies (references 11 and 12) based on chemical and cellular analyses of HHTAE. These studies highlight the low aristolochic acid content in HHTAE and suggest that the herb's toxicity is dependent on its alkaloid content. However, the authors conducted all experiments using crude HHTAE extracts. In my opinion, it would be valuable to perform in-vitro or in-vivo toxicity analyses using alkaloids only to better understand the role played by these compounds.
In the results section, the authors describe the outcomes of the "mouse experiment" with two single doses, 500 mg/kg and 2000 mg/kg. It is unclear how these doses were selected, and I recommend merging these sections since the observed effects are very similar.
The discussion section is brief, lacking the authors' hypotheses and comments to correlate the data obtained from zebrafish and mice. Given the disparate toxicity results between the two experiments, the authors should provide explanations for this discrepancy.
In the conclusion, the authors mention previously published data (references missing) without clearly specifying how these data correlate with the new findings.
Minor points:
- Page 1, lines 66-70: Correct and rephrase for clarity.
- Figures 4B, 5B, and 6B (tissue anatomy map) could be removed from the manuscript and placed in the Supplementary Materials. Additionally, these figures need modification to better illustrate the variance/invariance of mouse organs.
There are some editing and minor errors to address.
Author Response
Thank you for your valuable advice to improve the manuscript. We have revised the manuscript according to your suggestions. And the main revision and answers were listed in the attached file.

Reviewer 2 Report
Comments and Suggestions for Authors
The manuscript submitted for review presents experiments on zebrafish and mice to demonstrate the potential toxicity of alkaloids extracted from Hoittuyniae Herba. The topic is interesting, but the study raises some doubts.
Why was the toxicity of the analyzed compounds (aristolochic acids and aristolactams) not analyzed in the previous work, at least on zebrafish?
In what range of HHTAE concentrations has toxicity been tested and why these?
What was the group size of the zebrafish study, and the number of experiment repetitions?
Extraction from 30 kg of material suggests that this process was performed only once. Why? What is the efficiency of the extraction process?
On what basis were the administered doses to animals (e.g. 500 or 1000 mg/kg) determined?
Fig.1. – What do the dots in the diagram mean?
It is necessary to expand the discussion of the results. Currently, the Conclusions chapter is longer than the Discussion.
The consumed plant is a mixture of various substances in variable proportions. The effect of strengthening or weakening the effect (also toxic) of individual substances present in the plant is possible. It would be valuable to supplement these studies with analogous ones conducted on the mixture (alkaloidy & aristolochic acids and aristolactams AND alkaloidy & aristolochic acids and aristolactams & flawonoidy, itp.). I believe that the obtained results then will be more reliable and will significantly increase the value of this work.
Moreover, instead of experiments on mice (especially such a large number of mice), it may be possible to conduct tests on lower organisms, such as shrimp or daphnia, and/or cell lines. It's a pity that so many mice were killed, and the obtained results are not such significant.
Author Response

(The authors gave the same response as above.)

Round 2
Reviewer 1 Report
Comments and Suggestions for Authors
The authors have revised the manuscript according to the reviewer’s criticisms. However, the discussion section needs to be expanded and revised according to the following considerations:
Solubility of aristolactam components: The text mentions the poor solubility of aristolactam components. This limitation could impact the ability to conduct experiments over a wide range of concentrations, affecting the precision and reliability of the results.
Use of a single extract: The in vivo experiment were conducted only on the HHTAE extract due to the insufficient quantities of aristolactam compounds. This limitation may raise questions about the representativeness of results obtained from a single extract.
Differences between animal models: The text acknowledges differences in results between zebrafish and mice. This raises questions about the transferability of results from one animal model to another and emphasizes the need for further investigations to fully understand toxic effects.
Comments on the Quality of English Language
The English language in the discussion section requires moderate editing.
Author Response
Thank you for your valuable advice to improve the manuscript. We have expanded the discussion section according to detailed suggestions. The detailed information was highlighted in “Section 3. Discussion” of the manuscript.
Reviewer 2 Report
Comments and Suggestions for Authors
After correction, the work looks much better, and can be published. Personally, I am not convinced to perform experiments on animals if they are not necessary. In this case, there are still some problems to solve.
Author Response
Thank you for your advice to the manuscript. We will keep working on the toxicity research of aristolactams.